# Assistive Technologies for Supporting the Wellbeing of Older Adults

Ioanna Dratsiou [1,*], Annita Varella [1], Evangelia Romanopoulou [1], Oscar Villacañas [2], Sara Cooper [3], Pavlos Isaris [4], Manex Serras [5], Luis Unzueta [5], Tatiana Silva [6], Alexia Zurkuhlen [7], Malcolm MacLachlan [8] and Panagiotis D. Bamidis [1]

1   Medical Physics and Digital Innovation Laboratory, School of Medicine, Faculty of Health Sciences, Aristotle University of Thessaloniki (AUTH), 541 24 Thessaloniki, Greece; varellaannita@gmail.com (A.V.); evangeliaromanopoulou@gmail.com (E.R.); pdbamidis@gmail.com (P.D.B.)
2   Clinica Humana, 07010 Palma de Mallorca, Spain; ovillacanas@clinicahumana.es
3   PAL Robotics, 08005 Barcelona, Spain; sara.cooper@pal-robotics.com
4   Science For You (SciFY) PNPC, 153 41 Athens, Greece; paul@scify.org
5   Vicomtech Foundation Basque Research and Technology Alliance (BRTA), 48009 Bilbao, Spain; mserras@vicomtech.org (M.S.); lunzueta@vicomtech.org (L.U.)
6   Tree Technology SA, 28223 Madrid, Spain; tatiana.silva@treetk.com
7   Gewi-Institute for Healthcare Studies, 50678 Koln, Germany; zurkuhlen@health-region.de
8   Assisting Living & Learning (ALL) Institute, Maynooth University, Mariavilla, W23 F2K8 Maynooth, Co. Kildare, Ireland; mac.maclachlan@mu.ie
*   Correspondence: idratsiou@gmail.com

**Abstract:** As people age, they are more likely to develop multiple chronic diseases and experience a decline in some of their physical and cognitive functions, leading to the decrease in their ability to live independently. Innovative technology-based interventions tailored to older adults' functional levels and focused on healthy lifestyles are considered imperative. This work proposed a framework of active and healthy ageing through the integration of a broad spectrum of digital solutions into an open Pan-European technological platform in the context of the SHAPES project, an EU-funded innovation action. In conclusion, the SHAPES project can potentially engage older adults in a holistic technological ecosystem and, therefore, facilitate the maintenance of a high-quality standard of life.

**Keywords:** older adults; healthy ageing; assistive technology; cognitive training; physical training; social robots; voice assistant; chatbot; emotion recognition



## 1. Introduction

People throughout the world are ageing [1], and as they age, they are more prone to develop multiple chronic diseases and related impairments [2], as well as experience an inevitable decline in their physical and cognitive functions [3]. Over the years, biological ageing may affect older adults' independent living, and, as a result, many of them have limited opportunities to participate in active living and struggle with communicating, concentrating, memorizing, talking, walking, or maintaining their balance. In addition, ageing-related cognitive decline may greatly impact older adults' daily life [4], and this state of vulnerability can detrimentally undermine the process of active ageing, as well as the older adults' wellbeing [5]. In this vein, effective interventional strategies and innovative solutions that empower older adults' physical and cognitive function and promote their active and healthy aging are considered essential as they may have great benefits not only for older adults, but also for their caregivers and the whole society [6].

The use of Information and Communication Technology (ICT) and Assistive Technologies (ATs) aim to maintain and even enhance older adults' cognitive and physical condition, sense of security and safety, as well as their overall quality of life [7]. ICT's and AT's use in the health domain is progressively expanding [8], including a diverse array of devices,

services, strategies, and practices addressing the alleviation of the adversities that older adults face in their everyday living [7]. Specifically, they are designed to simplify older adults' lives and improve their ability to accomplish complex tasks in an effective way, through compensating acquired impairments that might emerge as a consequence of a wide variety of neurodegenerative diseases [9].

In particular, various studies exploring the effectiveness of technology-based interventions in providing cognitive, physical, and emotional support for older adults and their caregivers are reported in literature [5,10–13]. Computerized Cognitive Training (CCT) programs have emerged as a promising strategy to promote healthy cognitive ageing by providing older adults access to gamified, attractive, cognitive exercises everywhere and at any time [14]. Participation and systematic engagement in CCT interventions can potentially improve older adults' overall function in areas including memory, attention, speed, executive functions, orientation, and cognitive skills [15–17], while a combination of physical activity with cognitive stimulation may delay cognitive decline in older adults with dementia [15]. In addition to CCT interventions, there is an additional emergence in Natural Language Processing (NLP), which enables text analysis, extraction, and the automatic summarization of useful information, helping older adults to manage and minimize the volume and complexity of the information they receive.

Furthermore, there is an increased awareness on the potential role of robots in supporting older adults in maintaining their independence and wellbeing. Healthcare, rehabilitation, and social robots are among the many types and applications available [18]. Following this perspective, social robots, i.e., robots that engage in social interactions with their human partners, and often elicit social connections in return, have also been investigated, particularly in dementia care [19]. These types of robots act as companions for older adults and have the potential to benefit older adults' mental and psychological well-being [20,21]. Robots have indeed been found to positively affect engagement and motivation in the context of cognitive training, offering personalized care and therapy effectiveness [22]. Given this, robots should therefore be able to adapt to, develop, and recall a clear understanding of the means of human communication, thereby ensuring a high level of effective Human–Robot Interaction (HRI) [23]. This could be achieved through the integration of speech, face, and emotion recognition systems into the robot. In particular, speech recognition is enabled through the integration of Voice Assistants, which are interactive human–machine interfaces that allow natural and frictionless communication [24]. Due to their naturalness and ease of use, the interest in Voice Assistants in social and clinical domains has grown in recent years [25]. However, the development of Voice Assistants has emerged as a challenging task, where a pipeline of highly specific technological modules is built to have a more efficient interaction with the user [26]. In parallel, face recognition is achieved through the deployment of modern Deep Neural Network (DNN)-based face recognition solutions [27]. Face recognition is considered to be an attractive feature for older adults, as user authentication is enabled without the repetitive input of usernames and passwords, while, at the same time, a one-time login provides them with access to a full Internet-of-Things (IoT) network of different devices and/or robots [28]. Furthermore, emotions are considered to play an integral role in human communication [29] and, thus, emotion recognition is highlighted as a key component of affective computing [23]. Emotion recognition has increasingly attracted researchers' interest from diverse fields, and both the interpretation and understanding of human emotional states have been underlined as essential components in the HRI [30].

This study presents a broad spectrum of digital technologies integrated into a healthcare technological ecosystem for active and healthy ageing, facilitating the provision of more integrated services and care to older adults and supporting the maintenance of a higher-quality standard of life.

## 2. Shaping the Framework of Active and Healthy Ageing

### 2.1. The SHAPES Project

The SHAPES project [31] aims to create the first European open ecosystem enabling the large-scale deployment of a broad range of digital solutions for supporting and extending healthy and independent living for older adults, who are facing permanently or temporarily reduced cognitive functionalities and physical capabilities. The development of a large-scale, EU-standardized open platform will provide a broad range of interoperable technologies to improve the health, wellbeing, and independence of older adults, while enhancing the long-term sustainability of health and care systems in Europe. The SHAPES Pan-European Piloting Campaign will potentially engage more than 2000 older adults in fifteen (15) pilot sites, including six (6) European Innovation Partnership on Active and Healthy Ageing (EIP on AHA) Reference Sites [32], and involving hundreds of key stakeholders, such as older adults, their families, caregivers, and care service providers.

### 2.2. SHAPES Methodology

#### 2.2.1. Co-Design Process to Develop Personas, Scenarios, and Use Cases

Seven (7) Pilot Themes are being explored within the SHAPES Pilot Campaign reflecting a common framework to the different small-scale and large-scale pilot activities. Based on the pilot methodology, a human-centered co-design process was followed to de-sign and develop the different pilot activities and persona-based use cases under each Pi-lot Theme. A three-stage methodology was adopted: Creation of personas; Development of persona-based use cases; and Refinement of specific use cases of Pilot Themes.

As a first step, the personas' creation was actualized on the basis of mini-ethnographic studies and in-depth interviews with experts and focus groups designed and conducted by SHAPES partners. Through this process, the living context of older individuals was explored, leading to the design of the persona-based use cases under each Pilot Theme. As a second step, the persona-based use cases were generally envisioned as user stories, with storylines depicting users' activities and decisions in a specific context, while the range and variety of the devices, applications, and technological solutions within the living environment of older adults were identified. The third step was the refinement of specific use cases of Pilot Themes designating the main criteria for the selection of effective personalized technological solutions contributing to older adults' improved integrated care. Three main sources were used to determine the most suitable components and technological solutions for each of the use cases:

a.  Technological solutions included in available scientific literature;
b.  Technological solutions co-designed and co-executed by SHAPES partners with significant expertise in health and care settings;
c.  Third-party technological solutions from the Open Calls.

Conclusively, twenty-four (24) revised and improved use cases emerged based on the collaborative evaluation and agreement by the SHAPES partners and were allocated accordingly across the seven (7) Pilot Themes.

#### 2.2.2. SHAPES Pilot Campaign

The SHAPES Pilot Campaign follows a stepwise incremental approach, including: (1) Design and Preparation and (2) Deployment and Execution. The first part comprises the description of pilot objectives, needs, and KPIs and includes the development of: Table-top Exercises to validate initial concepts and approaches; Mock-up or Prototype Validation to assess technology acceptance and feedback on user experience; and Hands-on Experiments to validate functional elements of SHAPES components and digital technologies. The second part encompasses deployment and co-experimentation cycles in real-life environments; small-scale demonstrations to experiment with SHAPES digital technologies in a controlled environment; large-scale Pan-European piloting activities to test and validate the SHAPES technological platform.

Two use cases are discussed in the study: one from Pilot Theme 2 "Improving In-Home and Community-Based Care" and one from Pilot Theme 4 "Psycho-social and Cognitive Stimulation Promoting Wellbeing". These use cases are currently undergoing the Hands-on Experiments validation, involving key stakeholders, to validate the digital technologies' functionalities and provide feedback on user experience and other nonfunctional elements, including accessibility and inclusion features, necessary amendments, and improvements.

## 3. Deploying the Emerging Use Cases within Pilot Themes

### 3.1. Improving In-Home and Community-Based Care

The "Improving In-Home and Community-based Care Services" Pilot Theme is focused on providing an appropriate in-home setting for older adults mostly living on their own, who are in need of continuous healthcare provision, due to their suffering from permanent or temporarily reduced functions or capabilities, resulting from chronic age-related illnesses or declines. The pilot activities will build a safe and caring environment at-home that aim to promote older adults' autonomy and support through technology, and contribute to the long-term sustainability of health and care delivery for them. It will be implemented in rural as well as urban areas and, in particular, in the pilot sites Gewi-Institut für Gesundheitswirtschaft e.V. in Germany and in the Thessaloniki Active and Healthy Ageing Living Lab (Thess-AHALL) [33] in Greece.

This Pilot Theme encompasses four use cases that aim to build the multidimensional support of older adults through the exploitation of different digital technologies: remote monitoring of key health and wellbeing parameters, support interaction with the community, engagement in cognitive and physical training and provision of night surveillance rounds at the community care level in the home setting. The SHAPES digital technologies deployed in this Pilot Theme refer to an IoT-based living platform and web-based communication tool, monitoring devices, cognitive and physical training platforms and applications, as well as a social robot.

"LLM Care Health and Social Care Ecosystem for Cognitive and Physical training" is one of the use cases that are being deployed within this Pilot Theme and is, particularly, explored in this study. Through this use case, the integrated home-based social and health care service is provided by addressing the maintenance of cognitive and physical condition along with the promotion of independent living and healthy ageing of older adults. The target group of this use case is older adults with or without neurodegenerative diseases, mild cognitive impairment and mild dementia, chronic and mental disorders (e.g., schizophrenia), and difficulties in the production or comprehension of speech (e.g., tracheostomy, aphasia, stroke, and brain injuries). The Integrated Health and Social Care System Long Lasting Memories Care—LLM Care acts as an umbrella for the digital technologies of Talk & Play, Talk & Play Marketplace, and NewSum that are included in this use case and described below.

#### 3.1.1. The Long-Lasting Memories Care—LLM Care

The Long Lasting Memories Care—LLM Care [34] is an integrated ICT platform that combines cognitive training exercises [35] with physical activity [36], providing evidence-based interventions in order to improve both cognitive functions and overall physical condition [37], as well as quality of life. The combination of cognitive and physical training provides an effective protection against cognitive decline as age-related, thus improving the overall quality of life through the enhancement of physical condition and mental health, while preventing any deterioration and social exclusion [16]. LLM Care is considered a nonpharmaceutical intervention against cognitive deterioration that provides vital training to people belonging to vulnerable groups in order to improve their mental abilities while simultaneously boosting their physical wellbeing through daily monitoring. It has been recognized as an innovative ecosystem and was awarded a Transnational "Reference Point 2 *" within the EIP on AHA [37,38] due to its excellence in developing, adopting, and scaling up of innovative practices on active and healthy ageing.

LLM Care incorporates two interoperable components, physical and cognitive training. The physical training component, webFitForAll, is an exergaming platform (Figure 1) that was developed by the research group of Medical Physics and Digital Innovation Laboratory of Aristotle University of Thessaloniki within the European project "Long-Lasting Memories (LLM)" [34]. webFitForAll is addressed to older adults as well as individuals belonging to other vulnerable groups with the aim to promote a healthier and more independent living. It is based on new technologies and offers essential physical training within an engaging game environment through the incorporation of different gaming exercises (exergames) including aerobic, muscle flexibility, endurance, and balance training. More specifically, physical training is based on exercise protocols that have been proven to strengthen the body and enhance aerobic capacity, flexibility, and balance [39], while the adjustment of the difficulty level according to individuals' performance is provided aiming at achieving the optimum exercise performance.

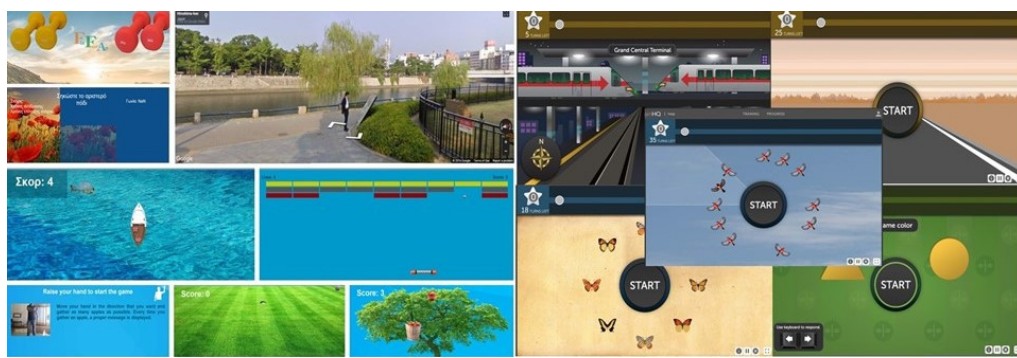

**Figure 1.** LLM Care physical and cognitive training.

The cognitive training component (Figure 1) is the specialized software BrainHQ that was designed and developed by Posit Science [35], in order to support cognitive game-based exercises in a fully personalized and adaptable cognitive training environment. Provision of personalized training, where each exercise is automatically adjusted to the beneficiary's level of competence, has been proven to accelerate and promote visual as well as auditory processing by improving memory, thinking, observation, and concentration [40]. BrainHQ is an online interactive environment that incorporates empowering cognitive techniques and includes six categories with more than 29 exercises with a hundred levels of difficulty, which focus on attention, memory, brain speed, people skills, navigation, and intelligence. It is addressed to older adults, as well as individuals belonging to other vulnerable groups aiming at a healthier and more independent living.

### 3.1.2. Talk & Play Desktop-Based Application

Talk & Play is a desktop application addressed to people with difficulties in the production or comprehension of speech (e.g., tracheostomy, aphasia, stroke, and brain injuries) with the aim to ensure sufficient communication and enhanced independence during their leisure time, while offering in-home cognitive training with the support of their caregivers [41]. Three categories of activities are provided: (a) Communication, (b) Games, and (c) Entertainment (Figure 2). In the context of the SHAPES Pilot Campaign, Talk & Play will be exploited among older adults and will be available in Greek, English, and German.

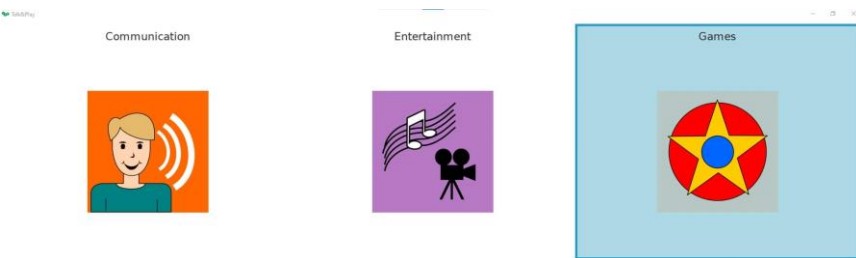

**Figure 2.** Talk & Play application interface.

In addition to that, the Talk & Play Marketplace will be provided as an expansion of the Talk & Play app acting as an online platform addressed both to formal and informal caregivers, providing them the opportunity to access, create, share, and download material and resources used in the Talk & Play app (Figure 3). More specifically, free access to a variety of communication and game resources will be provided, based on the crowdsourcing paradigm that promotes the generation of information, co-production of services, and creation of new solutions and public policies [42]. Caregivers will be able to explore and evaluate the content of Communication and Games categories and resources, as well as download the existing material and upload their own.

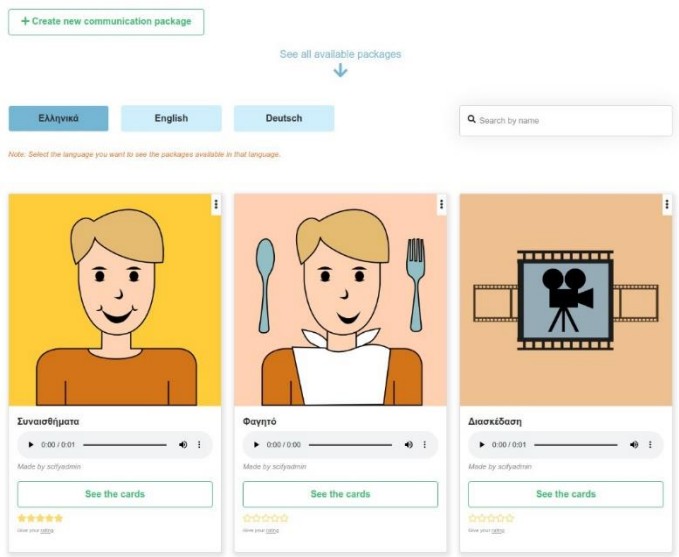

**Figure 3.** Talk & Play Marketplace interface.

### 3.1.3. NewSum App

NewSum [43,44] is a mobile application that enables news summarization based on Natural Language Processing (NLP) and Artificial Intelligence (AI). The NewSum app exploits state-of-the-art, language-agnostic methods of NLP to automatically develop news' summaries from a variety of news resources. Essentially, articles that refer to the same news' field are grouped in specific categories and concise summaries are created, without allowing the duplication of repeated information [45]. NewSum provides a simple User Interface (UI) design and its exploitation does not require a high level of digital skills and affinity to technology by users (Figure 4). Therefore, it can be exploited as a useful news summarization mobile app appropriate for older adults with little to no experience with technology and mobile apps. In the context of the SHAPES Pilot Campaign, NewSum will be available in Greek and English.

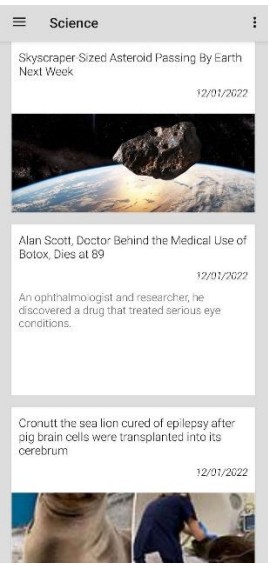

**Figure 4.** NewSum application interface.

### 3.2. Psycho-Social and Cognitive Stimulation Promoting Wellbeing

The Psycho-social and Cognitive Stimulation Promoting Wellbeing Pilot Theme is dedicated to technology-based interventions in living space environments for the improvement of the psycho-social and cognitive wellbeing of older adults, including those living with permanent or temporary reduced functions or capabilities. This Pilot Theme aims at using SHAPES psycho-social and cognitive gaming applications to make a positive impact on the older adults' healthy lifestyle and quality of life. It will be implemented in the pilot sites Clinica Humana (CH) in Spain, University College Cork (UCC) in Ireland, Thess-AHALL in Greece, and Italian Association for Spastic Care (AIAS) in Italy.

This Pilot Theme encompasses two use cases that aim to promote older people's physical and cognitive functioning with a particular focus on their psycho-social functioning, as well as supporting people with early-stage dementia through personalized cognitive activities. The SHAPES digital technologies deployed in this Pilot Theme refer to physical and cognitive training software, motion detection devices, emotion and face recognition software, chatbots, and social robots.

"A robot assistant in cognitive activities for people with early-stage dementia" is one of the use cases that are being deployed within this Pilot Theme. The target group of this use case is people with early-stage dementia, who are considered independent or highly independent and are in need of frequent assistance and support from either formal or informal caregivers. Through this use case, individual cognitive activities will be promoted to older adults based on the interaction with a social humanoid robot, which, among other features, will offer different interaction modes (text, sound, touchscreen, and voice recognition) depending on users' unique needs and preferences. The system will be connected to a panel where healthcare professionals and caregivers will be able to set-up the activities for older adults. As part of the SHAPES Pilot Campaign, a number of digital technologies will be integrated into the social robot ARI to increase its interaction capabilities: the DiAnoia and Memor-i cognitive activities; the Adilib chatbot, including speech interaction and reminders; the FACECOG face recognition module, providing encrypted authentication, and the emotion recognition module to detect user engagement.

#### 3.2.1. ARI Robot

ARI, the social robot (Figure 5), developed by PAL Robotics [46,47], has a human-like shape, autonomous behavior, and advanced HRI that enables the efficient display of the robot's intentions, the enhancement of human understanding, and the overall increase in users' engagement and acceptability [48].

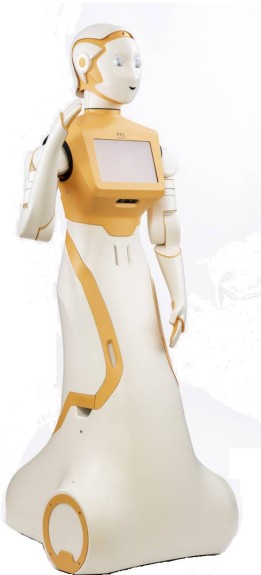

**Figure 5.** PAL Robotics ARI social robot.

ARI has a set of multi-modal behavior capabilities to increase users' acceptability and enrich the level of interaction. It incorporates various features including speech interaction to support multiple languages (Acapela Group's text-to-speech and Google Cloud API) and expressive animated eyes combined with head movement, so as to simulate gaze behavior. Default behavioral cues (waving, shaking hands, and nodding) and lifelike imitations (Alive module) motivate users to actively interact with ARI. Animated LED effects in both ears and the back torso may indicate both internal status of the robot (e.g., battery level) and interaction status (e.g., listening, waiting). ARI incorporates a touchscreen that allows the display of web-based content (images, videos, or HTML5 content) and includes both a temperature monitoring and alert system based on an input retrieved by the thermal camera located on the robot's head (Figure 6).

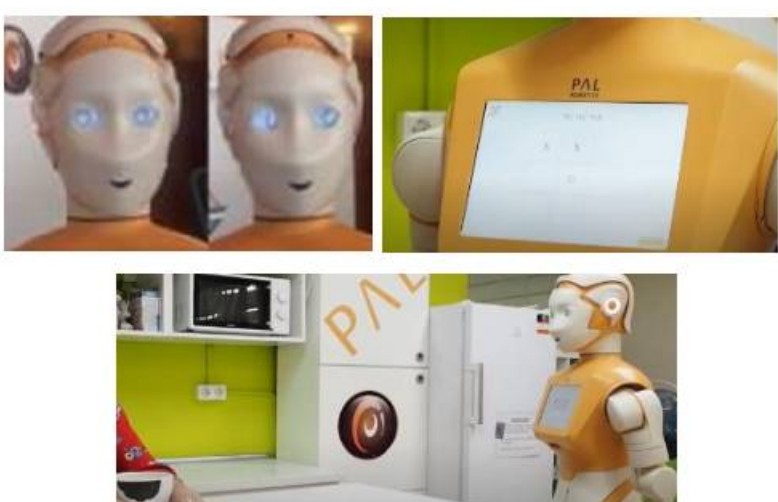

**Figure 6.** Different interaction modalities for the ARI robot.

Nodes of interaction are depicted in the robot's touchscreen, while text-to-speech and gestures are used to interpret them to users. In addition to the front touchscreen, an Android tablet located at the back of the robot can be used. Additional modifications include an RGB-D Intel RealSense camera on the head instead of the original RGB camera for more robust authentication, as well as a thermal camera for temperature monitoring.

Various features and functionalities will be incorporated in ARI, including autonomous navigation to specific rooms and the detection of pre-registered older adults in order to engage them in cognitive games (DiAnoia and Memor-i). Personalization in both cognitive games and the robot's behavior will be achieved based on the recognized user, using face recognition (FACECOG) and user engagement (Emotion recognition module). While the robot's multi-modal interaction capabilities will ensure the provision of feedback (e.g., speech, touchscreen displays, and LEDs) on users' performance and speech recognition, the chatbot (Adilib Chatbot) will enable the efficient interaction between users and the robot.

### 3.2.2. DiAnoia and Memor-i

DiAnoia [49] is a mobile app that provides nonpharmaceutical cognitive training to people with mild dementia and/or cognitive impairments focusing on attention, memory, thinking and communication, and executive functions. The app is primarily addressed to healthcare professionals and formal and informal caregivers of people with dementia, providing access to appropriate material for the cognitive training of their care recipients. In addition, Memor-i is an online free game, based on the "Memory Game" [50], that focuses on enhancing memory skills through pairing images with identical content (Figure 7). Memor-i is designed for people or/and older adults with visual impairments, who would preferably engage with an audio-based game as it offers binaural sound (3D audio), ensuring accessibility [51]. Interaction with Memor-i can be succeeded through the use of a keyboard, a mouse, or even a touchscreen in order to pair the identical images, while, as the game progresses, the difficulty level increases along with the number of the cards on the grid. Memor-i will be available in Greek, English, and Spanish.

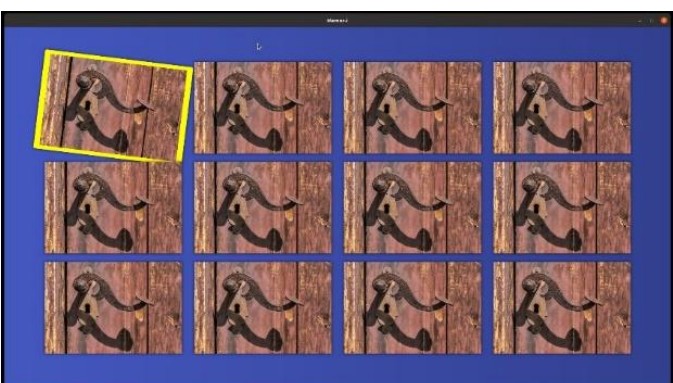

**Figure 7.** Memor-i desktop application.

In addition, DiAnoia Marketplace and Memor-i Studio will be provided, acting as online free apps targeted to multiple stakeholders (healthcare professionals, caregivers, family members, etc.) who support people with mild dementia and/or cognitive impairments and visual impairments (Figures 8 and 9). Following the crowdsourcing paradigm [42], both apps will provide users the opportunity to access various activities and games developed by related stakeholders, as well as create and download the content they develop. Specifically, users will be prompted to navigate through a simple UI and upload content, following the guidelines and requirements provided. Before publishing in the DiAnoia and Memor-i apps, each game and activity will be reviewed by the administrative team, in order to ensure that all necessary requirements have been thoroughly applied.

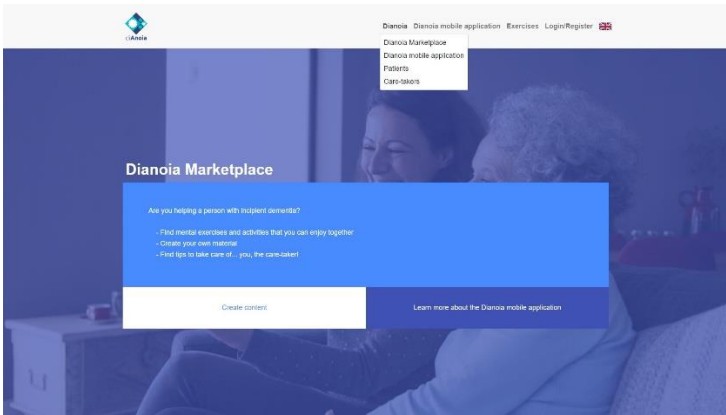

**Figure 8.** DiAnoia Marketplace.

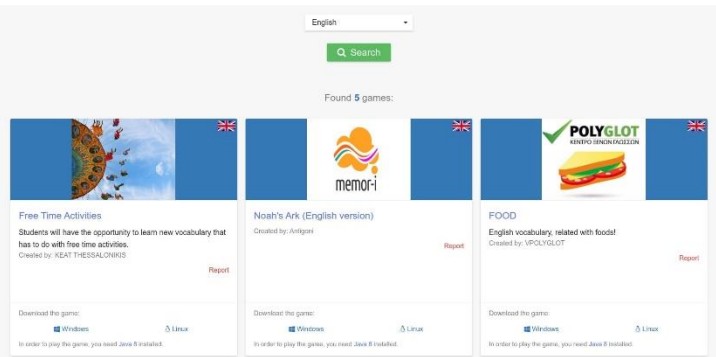

**Figure 9.** Memor-i Studio web application.

### 3.2.3. FACECOG—Face Recognition Module

The SHAPES Pilot Campaign will be conducted through the use of a large-scale open technological platform consisting of a broad range of heterogeneous IoT platforms, digital technologies, and services. In light of this, older adults might be confronted with difficulties during their interaction with the technological platform and, therefore, need guidance and support from their caregivers. FACECOG (Face Recognition Solution for Heterogeneous IoT Platforms), a software module from Vicomtech's Viulib library [52], provides and supports a password-based user authentication process, ensuring a friendly UI as well as the recognition of potential users at a distance (Figure 10).

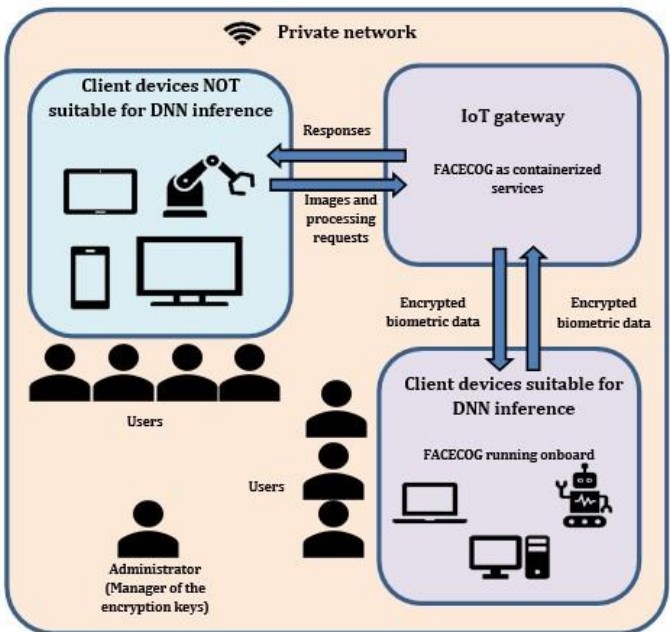

**Figure 10.** Deployment diagram of FACECOG.

FACECOG can be directly deployed on devices and robots with sufficient computing capabilities, such as ARI, to reduce the latency as much as possible in all cases. To allow users to register in one device and be recognized by another, the biometric data should be shared among the devices where it is deployed. Even if the network is private, the data would be encrypted, and an administrator would manage the encryption keys to preserve privacy with a secure element. In order to avoid compromising the privacy of users, FACECOG includes an encryption mechanism, based on fully homomorphic encryption that allows maintaining the biometric data always encrypted, even during matching operations [53]. User registration and verification workflows are presented in Figure 11.

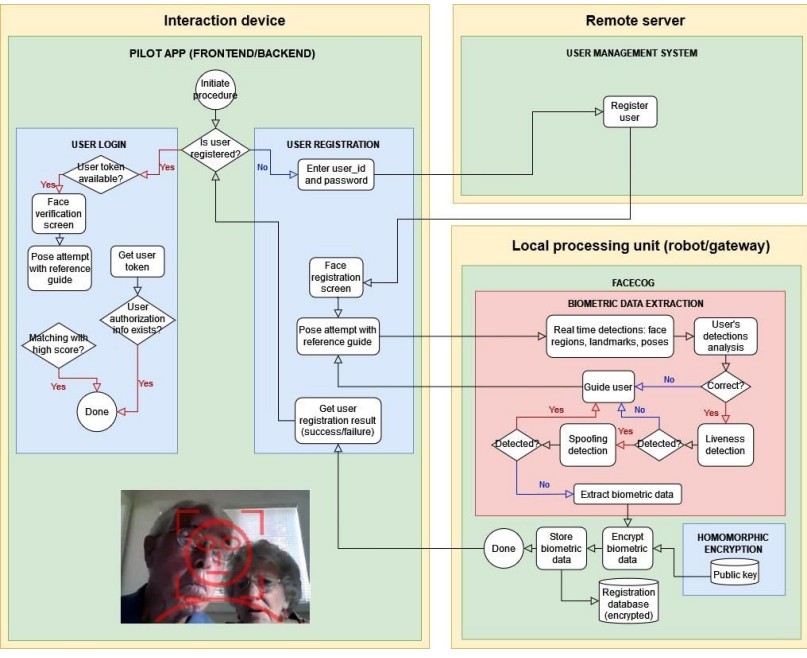

**Figure 11.** User registration workflow.

The high-quality extraction of biometric data is ensured based on the provision of visual feedback to the users. In particular, visual feedback is achieved through the notification of the tracked user and the subsequent provision of adequate guidance on the correct face position ("correct", "look straight", "move closer", "move backwards", "move to the right", "move to the left", "move downwards", and "move upwards"). When a high-quality facial image is captured, it is automatically registered in the system and the recognition process is enabled (Figure 12).

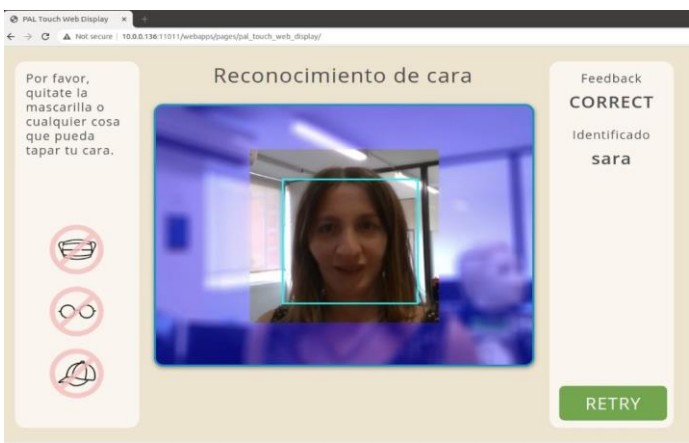

**Figure 12.** Face registration interface.

### 3.2.4. Emotion Recognition Module

The emotion recognition module developed by TREE [54] focuses on the integration of a set of digital technologies within the field of computer vision, exploiting the deep learning paradigm. Through the exploitation of cameras embedded in the ARI robot, an initial determination of the different emotional states of older adults during the interaction with ARI is achieved, detecting up to eight different emotions: neutral, happy, sad, surprise, fear, disgust, anger, and contempt. Additional characteristics (eye ratio, gaze direction) are extracted to allow the calculation of a complex metric of the current user's engagement with the tasks performed in front of the camera.

### 3.2.5. Adlib Chabot

Building a Voice Assistant in the context of the SHAPES project and tailored to cover the needs of a variety of different use cases scenarios would be unfeasible and too costly. To this end, a skill-based architecture using Adilib is proposed. Adilib Dialogue Skills are templates that allow users to instantiate closed-logic interactions such as personal reminders or tutorials. In this skill-based setup, an Orchestrator based on Language Representation models [55] selects who has to handle the users' query, e.g., if the user asks "I want to check my appointments", the Orchestrator will route that interaction to the "Agenda" skill to give an appropriate response, as depicted in Figure 13.

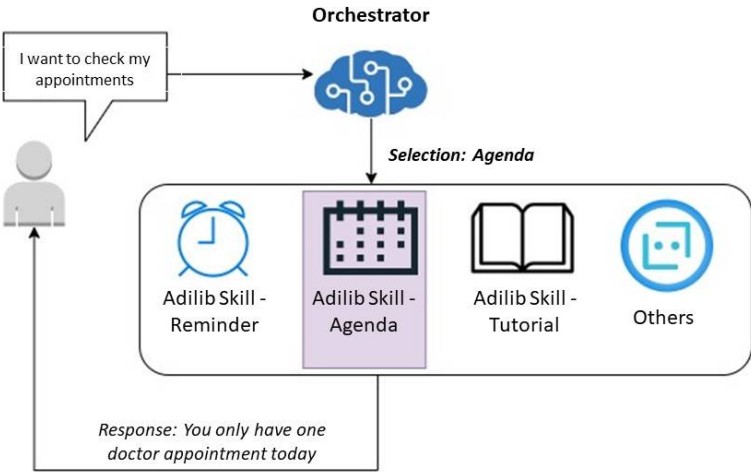

**Figure 13.** Deployment diagram of Adilib Chatbot.

Personalization of each Dialogue Skill's content responding to unique needs of older adults is provided through the use of a simple web UI, filling simple forms to adjust the dialogue template of the skill to their needs. The web form to add a doctor's appointment to the "Agenda Skill" is presented in Figure 14. Based on the information filled in this form, the Voice Assistant acts differently when presenting this agenda event to the user, e.g., if the escort field is defined in the web form, the assistant will be able to give a proper response to the question "Is anyone accompanying me there?" With this skill-based architecture, the interactions with the Voice Assistant are easily tailored and personalized even by caregivers without specific training on virtual assistant technical development.

**Assign Doctor Appointment**

| | |
|---|---|
| * Select a user from the list: | None ⇕ |

| Appointment information | Datetime and location information |
|---|---|
| * 1. Title | * 1. Appointment date and time |
| e.g, Monthly endocrine checkup | Format: DD/MM/YYYY-HH:MM |
| 2. Speciality | * 2. Timezone    3. Duration |
| e.g, Endocrine | Europe/Madrid (UTC + 2) ⇕   Unknown ⇕ |
| 3. Escort | * 4. Location |
| e.g, Your sister Julie | Enter the location of the appointment |
| * Mandatory fields | Save in the agenda |

**Figure 14.** Agenda Skill interface.

## 4. Discussion

Increased efficiency gains in integrated care policies and digital technologies for older adults across Europe are expected through the actualization of the SHAPES Pan-European Pilot Campaign. The aim of this study was to introduce a broad spectrum of digital technologies integrated into a standardized, interoperable, and scalable technological platform addressing active and healthy ageing and, therefore, supporting the maintenance of a higher-quality standard of life. As it has been already underlined, seven (7) Pilot Themes are being explored in the SHAPES Pilot Campaign, and two use cases from Pilot Theme 2 "Improving In-Home and Community Based Care" and Pilot Theme 4 "Psycho-social and Cognitive Stimulation Promoting Wellbeing" are the particular focus for this study.

More specifically, the ecosystem of the digital technologies and their integration in the use cases discussed in this study is conducted at the early and mid stages of the SHAPES project during the Design and Preparation stage. In this context, a three-phase process (Table-top Exercises, Mock-up or Prototype Validation, and Hands-on Experiments), including a series of validation actions involving relevant user and key stakeholders, is planned. The three-phase process is considered vital in emphasizing the critical factors for the successful planning and organization of the SHAPES pilot activities. Findings of the Design and Preparation stage will contribute in setting the basis for a first prototype of the digital technologies to be implemented, will inform specific requirements for the SHAPES technological platform, as well as will improve recommendations and amendments in general. Meanwhile, the usability and feasibility tests will provide technical partners with essential feedback on the proposed integration of the digital technologies deployed in both use cases and, therefore, act as a future basis for further planning of the pilot activities.

Future steps will focus on the Deployment and Execution stage, which will address the incremental deployment of the platform's instances across the different European pilot sites through co-experimentation cycles in real-life environments. Following this perspective, small and large-scale piloting activities will be conducted, aiming to incrementally inform the platform's development, as well as experiment and test all SHAPES digital technologies. Specifically, during the small-scale piloting activities, the feasibility of methods and procedures for later exploitation on the large-scale Pilot Campaign will be tested. This small-scale demonstration will be performed with a smaller group of participants and/or with fewer SHAPES digital technologies to identify factors that could hinder the pilot site to organize and perform a successful large-scale demonstration. At a later stage, the large-scale piloting activities will test all digital technologies, methods, and processes under real-life conditions among the targeted end-users in fifteen (15) European reference sites. The implementation of the SHAPES large-scale Pan-European Pilot Campaign will allow for the validation of the SHAPES technological platform capabilities and benefits to care recipients, caregivers, and care service providers across different regions, cultures, and health and care organizational models. In addition, it will assess its impact on the support of healthy ageing and independent living and will define improved integrated care policies and measures.

At a higher level, outputs from both the small and large-scale SHAPES Pilot Campaign activities will aim to facilitate older adults' long-term healthy and active ageing, as well as maintain a high quality of life through the exploitation of a broad range of integrated technological, organizational, clinical, educational, and societal solutions that will be enabled by SHAPES digital technologies and the technological platform. To this end, exploration of the results of the Design and Preparation process is considered essential for laying the groundwork for the successful implementation of the SHAPES Pilot Campaign. Future directions of this study could focus on expanding the work utilized in the Design and Preparation process and elaborate on presenting the upcoming results from the Pilot Campaign in a future larger-scale study. The integration of technologies to address people's key challenges as they age is a key challenge of the World Health Organization's DATA (Digital and Assistive Technology for Aging) initiative [56].

To this end, SHAPES' vision focuses on creating an EU-standardized platform, incorporating and integrating a wide spectrum of digital technologies that will enable older adults living across European regions to remain healthy, active, and independent while maintaining a high quality of life for as long as possible.

## 5. Conclusions

Declines in older adults' physical and cognitive function may hinder their ability to perform daily tasks, and consequently, decrease their independency and their quality of life. There is great potential to introduce innovation and technology-based interventions in older adults' daily living, in accordance to their own unique needs and preferences. To this end, the SHAPES project's ambition is to integrate a broad range of digital technologies into a

healthcare Pan-European technological ecosystem that will support and extend the healthy, active, and independent living of older adults, through a large-scale Piloting Campaign in multiple and diverse areas across Europe.

**Author Contributions:** Conceptualization, I.D., A.V. and E.R.; writing—original draft preparation, I.D., A.V. and E.R.; writing—review and editing, O.V., S.C., P.I., M.S., L.U., T.S., A.Z. and M.M.; supervision, P.D.B.; project administration, M.M. All authors have read and agreed to the published version of the manuscript.

**Funding:** This research has been supported by the SHAPES Smart and Healthy Ageing through People Engaging in Supportive Systems funding from the European Union's Horizon 2020 research and innovation program under grant agreement No. 857159-SHAPES-H2020-SC1-FA-DTS-2018-2020.

**Institutional Review Board Statement:** Not applicable.

**Informed Consent Statement:** Not applicable.

**Data Availability Statement:** Not applicable.

**Acknowledgments:** The authors would like to thank the SHAPES consortium for its support and contribution.

**Conflicts of Interest:** The authors declare no conflict of interest.

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
