# Peer review of "Assistive Technologies for Supporting the Wellbeing of Older Adults"

_technologies, doi:10.3390/technologies10010008_

Round 1

Reviewer 1 Report

The study presents a set of potential solutions to adress age related challanges- mental cognitive and social however it does not provide criteria to assess these technologies compared to other existing technologies. For stakeholders, who are interested inimplementation this study does not provide any tools how to choose\develop technologies for the target population 

Reviewer 2 Report

The authors present a framework for the integration of a broad spectrum of digital solutions into an open Pan-European technological platform in the context of the SHAPES project, an EU-funded innovation action.

The paper is well written and organized.

However, the paper looks basically like a list of the integrated digital technologies. I suggest to add a more meaningful discussion on the expected issues that authors envisage for the next steps of the project and potential solutions to face the challenges.
